# Engineering Photosensory Modules of Non-Opsin-Based Optogenetic Actuators

**DOI:** 10.3390/ijms21186522

**Published:** 2020-09-07

**Authors:** Xiaocen Lu, Yi Shen, Robert E. Campbell

**Affiliations:** 1Department of Chemistry, University of Alberta, Edmonton, AB T6G 2G2, Canada; xiaocen@ualberta.ca (X.L.); yshen3@ualberta.ca (Y.S.); 2Department of Chemistry, The University of Tokyo, Tokyo 113-0033, Japan

**Keywords:** optogenetic actuator, protein engineering, photosensory domains

## Abstract

Optogenetic (photo-responsive) actuators engineered from photoreceptors are widely used in various applications to study cell biology and tissue physiology. In the toolkit of optogenetic actuators, the key building blocks are genetically encodable light-sensitive proteins. Currently, most optogenetic photosensory modules are engineered from naturally-occurring photoreceptor proteins from bacteria, fungi, and plants. There is a growing demand for novel photosensory domains with improved optical properties and light-induced responses to satisfy the needs of a wider variety of studies in biological sciences. In this review, we focus on progress towards engineering of non-opsin-based photosensory domains, and their representative applications in cell biology and physiology. We summarize current knowledge of engineering of light-sensitive proteins including light-oxygen-voltage-sensing domain (LOV), cryptochrome (CRY2), phytochrome (PhyB and BphP), and fluorescent protein (FP)-based photosensitive domains (Dronpa and PhoCl).

## 1. Introduction

Optogenetics, a set of technologies that involves the use of light and genetically encoded photo-responsive proteins to control cellular activities, has enabled a tremendous number of biological breakthroughs in recent years [1,2,3]. Compared to conventional synthetic organic photoswitches, optogenetic tools allow precise targeting to specific cell types or subcellular regions in a relatively non-invasive way. In addition, numerous emerging optogenetic tools have brought new insights into modern neurobiology and cell physiology by enabling fast and reversible control of transient activities [4]. This combination of light and genetic engineering provides substantial advantages towards the investigation of many biological processes, such as uncovering the mechanism of complex processes in the brain (e.g., learning [5], sleep [6], addiction [7], and movement [8]), and understanding the pathology of neurological disorders [9,10,11].

In a broad sense, optogenetic tools can be classified into two complementary groups: Light-dependent controllable actuators and light-dependent optical indicators [12]. Optogenetic actuators are proteins that can control a specific biological activity when illuminated, and optogenetic indicators are protein-based biosensors that can transduce microenvironment changes into detectable optical readouts. The combined use of spectrally and biologically compatible pairs of actuator and indicator allow simultaneous manipulation and monitoring of biological processes [13,14,15]. Although early optogenetics tools were primarily used in neuroscience, recent advances in the tool development area are addressing a wide scope of studies across all the research areas of biological sciences [12,16,17,18].

Optogenetic actuators provide a powerful method to manipulate various subcellular applications, including neuron excitability [13,15,19], protein translocation [20,21], organelle transport [22,23], gene expression [24,25,26], genome editing [18,27,28], and intracellular signal transduction [29,30,31,32]. One of the key components of the optogenetic tools are the photosensory proteins, most of which are engineered from natural photoreceptors. The optical properties and light-induced responses of these naturally-sourced proteins can be optimized for specific applications through the use of protein engineering.

Within the optogenetic tool repertoire, the most widely used actuators in neuroscience are photosensitive transmembrane proteins, which are engineered from animal and microbial opsins [33,34]. Opsin-based actuators make use of the photo-isomerization of the retinal-based cofactors to generate ion currents across the cellular membrane [35,36,37], or to modulate intracellular signal transduction [38,39] ((i) in Figure 1). The light-activated systems were originally designed for optical control of neural electrical activity through opsin-based tools [12,19,35,37,40,41,42,43]. In the past decade, these opsin-based tools have been extensively developed into a variety of variants, enabling optical control of the activities over a range of timescales. These variants also involve different transduction mechanisms, and can be excited by various wavelengths [12,33,34].

Although the advantages of opsin-based optogenetic tools have been widely recognized and exploited in neurobiology, the non-opsin-based actuators are now playing increasingly important roles in optical control of protein activity through simple and versatile strategies. In the review, we aim to provide a timely overview of the engineering of non-opsin-based photosensory proteins and their representative applications in subcellular levels.

## 2. Photosensory Modules in Optogenetic Actuators

Optogenetic actuators typically consist of two functional moieties: A photosensory module and a biofunctional module. The photosensory module generally contains a cofactor chromophore or a chromophore generated from autogenic post-translational modification of amino acids, which enables the molecule to absorb light. The absorbance of light typically leads to a photochemical reaction that is associated with a change in the protein structure or function. The biofunctional module is the part that acts as the bridge between the photosensory module and the biological system. The light-induced change photosensory module structurally or functionally alters biofunctional module, resulting in a modulated interaction with the biological system.

At the protein structural level, the two modules can exist in a single protein domain, as in microbial opsin-based actuators (such as channelrhodopsin-2 [35] and halorhodopsin [36]). However, it is more common for the two modules to be distinct domains which are genetically connected in a protein fusion, as for photoactivatable enzymes actuators (such as photoactivatable Rac1 [57] and photoactivatable CRISPR-Cas9 [18]). A variety of design strategies (Figure 1) have inspired researchers to develop diversified optogenetic tools by using protein engineering via both rational design and directed evolution.

Within the current actuator’s repertoire, most of the biofunctional modules are engineered from a relatively wide variety of target molecule-binding proteins [59,60,61,62], ion channels [63], or pumps [64]. However, the light-sensitive protein moieties that serve as the photosensory modules of optogenetic actuators are limited to just several engineered naturally sourced photoreceptor proteins, including animal and microbial rhodopsins; plant phytochromes, cryptochromes, and phototropins; bacterial phytochromes; and coral photo-transformable fluorescent proteins (PTFPs). According to different light-induced responses, these engineered optogenetic actuators can be divided into four categories [58,65] (Figure 1): (i) Opsin-based light-activatable channels and pumps [35,36,37,38], (ii) proteins that undergo light-induced dimerization [24,25,27,32,44,47,49,50,54], (iii) proteins that undergo allosteric conformational changes with light stimulation [20,57], and (iv) a protein that undergoes photocleavage [58].

In subsequent sections, we will focus on the detailed protein engineering of the non-opsin-based photosensory domains, as well as the most commonly used design strategies for actuators and representative applications in cell biology.

## 3. LOV Domain-Based Optogenetic Tools

The light-oxygen-voltage-sensing domain (LOV domain), which belongs to a subclass of the Per-Arnt-Sim (PAS) superfamily [66], was first identified as the photosensory module in *Arabidopsis* plant phototropin 1 [67,68]. Since their initial discovery, LOV domains have been found in a variety of bacterial, algal, fungal, and other plant species. The flavin cofactor chromophore (FMN or FAD) [69] absorbs blue light (wavelength, ~400–480 nm) resulting in the reversible formation of a covalent cysteinyl-flavin (Cys-flavin) adduct in the hydrophobic core [68,70]. The photoreduction reaction triggers a protein allosteric change through the highly variable N- or C-terminal (e.g., the Jα helix) extensions [71] which completes the transduction of the light into a biochemical signal [66,72].

In the optogenetic toolkit, the LOV domain has been utilized for optical control of the biofunctional modules through two general strategies [31]: (i) a single light-activatable protein or peptide that drives a signaling pathway [20,21,57,73], and (ii) light-induced protein–protein interactions that manipulate the subcellular localization of a protein or a protein activity [47,49,50,74]. We will put focus on examples of engineering of LOV-based actuators from the perspective of the photocycle, proteins with light-induced allosteric responses, and proteins with light-induced dimerization responses.

### 3.1. Engineering of LOV Photocycles

The kinetics of the LOV domain photocycle is dependent on the redox processes of the flavin chromophore [31,75], which has been studied extensively (Figure 2a). Compared to the light-activation step with a timescale of microseconds [76,77], the thermal recovery from the Cys-flavin adduct in the light state to the ground state in dark is very slow in LOV domains, and is therefore the rate-limiting step in the photocycle. The half-lives of this process vary from a few seconds to days depending on the particular LOV domain [68,70,78,79,80,81].

Early efforts to engineer the LOV domain were focused on the use of rational design to optimize the dark recovery kinetics. Studies revealed that the Cys-FMN adduct cleavage is assisted by a base-catalyzed deprotonation of the flavin N5 [82] (Figure 2b). The mechanism was found to involve a highly conserved glutamine residue [83] and surrounding side chains that interacted with the flavin and associated water molecules [84,85,86].

According to the cleavage mechanism, there are three general strategies used in previous studies to alter the dark recovery rate [31,87]: (i) Modification of the electrostatic environment of the chromophore to accelerate or decelerate the deprotonation of the flavin N5 of the adduct [83,87,88,89]; (ii) increase or decrease the solvent accessibility of the cofactor [87,90,91]; and (iii) alter the stability of the flavin through stacking or steric interactions [87,88,92,93]. Mutational analysis of the vicinal residues near the binding pocket were performed to investigate photocycle-tuning mutations. The results suggested that substitutions which accelerate deprotonation of N5, increase solvent accessibility or destabilize the light adduct would shorten the photocycle [83,87,88,89,93]. In contrast, the photocycle can be lengthened by decelerating the deprotonation, decreasing the solvent accessibility or stabilizing the adduct state [87,92,93]. Here, taking the *Avena sativa* phototropin 1 LOV2 (AsLOV2) (wild type: τ_FMN_ = 81 s) [88,94] as an example, we summarize the lifetimes of previously reported AsLOV2 variants (Table 1).

### 3.2. Engineering of LOV Domains with Light-Induced Allosteric Responses

In the AsLOV2 domain, the formation of Cys-FMN adduct disrupts the β-sheet in the core of the domain and results in partial unfolding of the C-terminal Jα helix [84,90,95,96] (Figure 3a). The activity of the biofunctional domain fused to the Jα helix can be modified by this light driven allosteric change ((iii) in Figure 1). In previous studies, Zayner et al., have reported the characterization of more than 100 variants of AsLOV2 with mutational analysis based on both photocycle kinetics and conformational changes [88,89]. Their results reveal that there is no obvious correlation between these two properties, which suggests the kinetics and the dynamic change of this system can be regulated respectively.

Based on the results from mutational analysis [89], the positions that contribute to larger conformational change are mostly located on the A’α helix near to the Jα helix and the interface between the Jα helix and the LOV core (Figure 3b). Substitutions to hydrophobic residues in this region could stabilize the Jα helix docking in the dark state [97]. For optogenetic applications, stability of Jα helix docking in the dark state can alter the affinity, and larger allosteric change could increase the dynamic change of light-dependent activation.

The mutations shown in Figure 3b have been harnessed to optimize AsLOV2-based optogenetic tools for applications in cell biology. For example, the mutations: Thr406Ala, Thr407Ala, Ile532Ala [97] (increasing the affinity of Jα helix docking); Val529Asn [98] (decreasing the affinity of Jα helix docking); and Val416Ile [94] (decreasing the dark recovery rate), were used to tune the equilibrium and kinetic parameters in “tunable, light-controlled interacting protein tags” (TULIPs) reported by Strickland et al. in 2012 [49] ((ii) in Figure 1). In the TULIPs system, a peptide epitope was fused to the C-terminus of the Jα helix and incorporated in helix docking by rational design, such that its function was caged in the dark state. “Uncaging” of the peptide upon illumination led to recruitment of the peptide binding protein (ePDZ), leading to induction of light-dependent protein translocation or other protein–protein interaction (PPI). TULIPs system with various mutated LOV2 variants have been successfully applied for optical control of mitogen activated protein kinase (MAPK) activation [49].

Another important category of optogenetic tools that utilized the light-dependent allosteric change of AsLOV2 are the “light-inducible nuclear localization signal” (LINuS) and the “light-inducible nuclear export system” (LEXY) reported by Niopek et al. [20,21] ((iii) in Figure 1). In these systems, the localization peptide and Jα helix were incorporated together by rational redesign, followed with screening for variants with low dark state activity and high protein translocation efficiency upon illumination. Although the development of these system required substantial efforts in protein engineering, this versatile strategy enables the light-dependent protein translocation in a simpler way by fusing the biofunctional domain to the LINuS or LEXY. In addition to optical control of gene expression by manipulating the localization of transcription factor [20,21], LINuS and LEXY systems have also been used in optogenetic regulation of cofilin-1 for controlling F-actin assembly [99], manipulating of the endogenous p53 protein levels [100], and optogenetic translocation of TEV protease for manipulating protein degradation [101].

Despite these and numerous other successful examples (such as PA-Rac1 [57], PACR [102], BACCS [103], and BLINK1 [104]) of optogenetic tool engineering using the AsLOV2 photosensory domain, rational improvements of the tools used in biological applications are limited by subtleties of allosteric regulation and can vary from case to case. Generally, development of AsLOV2-based tools requires substantial empirical efforts for optimization of the LOV domain, linkers, and the protein of interest.

### 3.3. Engineering of LOV Domains that Undergo Light-Dependent Dimerization

Another LOV-based optogenetic control strategy is to alter PPI via the light-induced protein dimerization of certain LOV variants. This strategy was used in AsLOV2-based photosensory modules including TULIPs system (as discussed above), and a photo-induced protein dissociation system LOVTRAP reported by Wang et al. in 2016 [50] ((ii) in Figure 1). To develop LOVTRAP, a small protein named Zdark (Zdk), which selectively binds to the dark state of AsLOV2, was generated by mRNA display screening. Upon illumination, the binding is disrupted by the conformational change of the LOV2 domain, resulting in the uncaging of biofunctional domain by freeing the protein to move to its site of action. The LOVTRAP system, and its recently reported derivative Z-lock [52], have been used to mediate activation of different signaling protein that control cell-edge protrusion (Vav2, Rac1, RhoA, cofilin, and αTAT) and represent a versatile approach for reversibly photo-caging and uncaging.

In the TULIPs and LOVTRAP systems, the interaction between the AsLOV2 core domain and its engineered binding partner (ePDZ or Zdk) mainly depends on the naturally allosteric change of Jα helix, and the light-dependent PPIs were introduced later via protein engineering. By contrast, as an appropriate naturally sourced candidate, the fungal photoreceptor Vivid (VVD) from *Neurospora crassa* undergoes light-induced homodimerization or heterodimerization upon blue light illumination [80]. VVD shares very similar photochemistry to AsLOV2 and its photocycle can similarly be altered via rational design based on the strategies discussed above. Accordingly, the dark state lifetime of engineered VVD variants can vary from tens of seconds to several days [94].

For the wild type VVD, the light-induced conformational change propagates to the N-terminal cap (N-Cap) which induces protein homodimerization [72]. More specifically, the side chain rotation of the conserved Gln182 upon light adduct formation favors the hydrogen bonding with the reduced flavin and initiates the movement of the Cys71 side chain to interact with the β-stand where Gln182 is located [72,105,106]. The rotation is propagated through rearrangement of the N-Cap, which exposes a hydrophobic cleft to facilitate homodimerization. Overall, the formation of the homodimer is driven mostly by the conformational changes of the N-latch and N-hinge regions [105,106,107] (Figure 3c).

Due to its light-dependent homodimerization, VVD is a suitable scaffold for optogenetic control of PPIs. However, the relatively low affinity of the wild-type homodimer (*K*_d_ = 13 μM) [105] is not sufficient to induce robust light-dependent PPI in the context of live cells. To date, relatively few mutations, such as Cys71Val and Asn56Lys, have been reported to improve the affinity in the light state (but not the dark state) [105,106]. Compared to Cys, the Val side chain has two methyl groups that points to two directions, mimicking two rotamers of the Cys side chain. One methyl occupies the positions of thiol group in the dark state, and the other methyl mimics the thiol of Cys71 in the light state. The side chain of Val residue predisposes the VVD conformation towards the light state, thereby facilitating the dimer formation. Accordingly, the variant with the Cys71Val mutation has an increased affinity (*K_d_* = 0.4 μM) [105]. The Asn56Lys mutation is located on the helix of N-Cap and was found to form a salt bridge with the Asp68 residue on the second VVD domain and thereby stabilize the dimer [24].

The VVD variant with Cys71Val and Asn56Lys mutations were reported to exhibit lower background signal and larger dynamic range than the wild type VVD in the “LightOn” gene expression system [24] ((ii) in Figure 1). In this system, the dimerization motif of a transcription factor is replaced with the light-inducible VVD. Recently, the wild type VVD has also been used to develop a light-activatable recombinase system named RecV reported by Yao et al. [108]. In this design, by fusing the split recombinase to VVD through a specific topology, the N- and C- portions of the recombinase would complement to each other in the correct orientation upon light-activation. The RecV toolkit enables two-photon-mediated optogenetic genome modification in a single targeted cell.

Due to the fact that VVD undergoes a homodimerization, this tool cannot selectively bring two different proteins together (i.e., heterodimerization). To increase its utility, Kawano et al. developed two distinct types of VVD variants named pMag (positive) and nMag (negative) [47] ((ii) in Figure 1). pMag was engineered by the introduction of positively charged residues (substitutions Ile52Arg and Met55Arg) in the N-Cap, and nMag was engineered by introduction of a negatively charged residue (substitution Ile52Asp) and a neutral mutation Met55Gly to prevent homodimerization (Figure 3c). Unlike the original homodimeric VVD, the two variants engage in electrostatic interactions instead of primarily hydrophobic contacts and exhibit robust blue light-dependent heterodimerization (with a 21-fold change in bioluminescence complementary assay upon illumination). Moreover, the switch-off kinetics and dimerization efficiencies were optimized by introduction of mutations that have been previously reported to alter the photocycle lifetime [94]. The Magnet toolkit was demonstrated to be useful for the optical control of phosphatidylinositol (PI) 3-kinase (PI3K) for generation of PI(3,4,5)P_3_. Overall, this system broadened the scope of optogenetic control by using VVD. In particular, light-induced heterodimerization provides a simple and versatile way to control the activity of split enzymes including Cre recombinase [109], CRISPR-Cas9 [18], T7 RNA polymerase [48] and intracellular antibodies (intrabodies) [110]. Notably, the VVD-based optogenetic tools have been reported to be temperature-sensitive [111,112] and stimulating cells with colder temperatures can also induce dimerization [113]. Accordingly, the pre-incubation of cells at the experimental temperature [47], and keeping the temperature constant during cell imaging, are important considerations for this system.

## 4. Cryptochromes-Based Optogenetic Tools

Another type of photosensitive protein with a flavin cofactor is cryptochrome 2 (CRY2). CRY2 can simultaneously undergo light-induced reversible homo-oligomerization, and heterodimerization with its binding partner (CIB1), upon blue light illumination [53] ((ii) in Figure 1). Similar to the photochemistry of LOV2 domain, the conformational change of CRY2 is initiated by the photoreduction of a FAD cofactor chromophore located in the N-terminal photolyase homology region (PHR) of CRY2. Although the CRY2 flavin photochemistry is not well-established, the observed photocycle of wild type *Arabidopsis* CRY2 is reversible within minutes, thus making it suitable for transient stimulation of many biological processes naturally.

### 4.1. Engineering of CRY2-CIB1 Pair That Undergoes Light-Dependent Heterodimerization

The utility of the CRY2-CIB1 pair was first demonstrated through the optical control of transcriptional factor Gal4 and Cre recombinase in mammalian cells in 2010 by Kennedy et al. [27]. The wild type full length CRY2 and CIB1 had some limitations as a tool, including low expression level and punctate nuclear localization when expressed in mammalian cells. To address these problems, Kennedy et al. optimized this system by using only the PHR domain of CRY2 (CRY2PHR) and a CIB1 C-terminal truncated version (CIBN). They also mutated the predicted nuclear localization signal in CRY2 and CIB1. The resulting CRY2PHR and CIBN variants expressed much better in cells and showed evenly distributed cytoplasm localization [27].

In 2016, Taslimi et al. reported a second generation CRY2-CIB system with smaller protein size and reduced interactions in the dark state [28]. The improved truncated variant CRY2(535) (residues 1–535) was demonstrated to have larger dynamic range and lower dark state self-association than CRY2PHR. To improve the versatility of the CRY2-CIB1 system, mutations adjacent to the flavin chromophore that alter the photocycle were identified via directed evolution based on a yeast two-hybrid screening. Compared to the dark recovery rate of wild type CRY2 (*t*_1/2_~16 min) [114,115], the variant with long-cycling mutation Leu348Phe has a slower rate (*t*_1/2_~24 min), and the variant with fast-cycling mutation Trp349Arg has a faster rate (*t*_1/2_~2.5 min) in a cell membrane recruitment assay [27,28].

The optimized CRY2-CIB system has been applied widely in optogenetic manipulations of PPI, to control gene expression [27,28], genome editing [27,28], cellular signaling pathways [116,117], lipid metabolism [58], cell death [30], and organelle transport [22]. Notably, by introducing the long-cycling mutation Leu348Phe from the second generation CRY2, in 2019 Godwin et al. reported CRY2-CIB-mediated light-activatable BAX system (OptoBAX) [118], which requires less frequency light illumination for activation of apoptosis compared to the original CRY2 version.

### 4.2. Engineering of CRY2 That Undergoes Light-Dependent Homo-Oligomerization

In addition to heterodimerization, the light-dependent homo-oligomerization of CRY2 has been harnessed in optogenetic strategies to control a wide range of biological processes [44,45,119,120]. Although the light-induced oligomerization could be used to sequester or inhibit the biofunctional module by formation of protein clusters [45,121], the visible light-induced clustering of CRY2 typically requires high local protein concentration [44] and depends on the oligomeric tendency of the fused protein [122].

In 2014, Taslimi et al. reported a new CRY2 variant name “CRY2olig” with enhanced homo-oligomerization [44]. The key mutation (Glu490Gly) of this improved variant was discovered through yeast two-hybrid screening. CRY2olig showed improved clustering rate with half-maximal clustering time in the range of tens of seconds after blue light illumination, while the dark dissociation of the larger CRY2olig clusters (*t*_1/2_ ~23 min) is slower than the wildtype CRY2 (*t*_1/2_ ~6 min) [45]. The utility of this light-dependent cluster formation was demonstrated by visualization of protein interactions, and transient inhibition of biofunctional modules through surrounding protein crowding effect. Another variant with improved cluster formation, designated “CRY2clust”, was reported by Park et al. in 2017 [120]. The improved function was achieved by adding a 9 residue C-terminal extension following CRY2PHR domain. The improved clustering ability of CRY2clust was demonstrated in a Ca^2+^-modulating optogenetic system (OptoSTIM1) [123] and optical control of protein kinase activity [121]. However, the mechanism by which the Glu490Gly mutation and C-terminal short peptide extension modulate the oligomerization remained unclear due to lack of structural knowledge of the CRY2 interaction interfaces, which has also limited rational efforts to engineer the CRY2 system.

To better understand the mechanisms of CRY2 interactions, in 2017 Duan et al. proposed the charged residues that affect the heterodimerization and homo-oligomerization based on a model of CRY2 obtained by homology modeling [124]. They proposed that the CRY2-CIBI and CRY2-CRY2 interactions are dependent on different binding sites on CRY2. The N-terminus (residues 2–6) of CRY2 is exposed to solvent, and is important to the heterodimerization to CIB1. The N-terminus (containing three Lys residues and one Asp residue) is highly charged and has a net positive charge at physiological pH. The deletion of the N-terminus or mutations of the charged residues to neutral residues significantly reduces the light-dependent binding affinity to CIB1 without impacting homo-oligomerization. In contrast, C-terminal residues tend to be crucial for CRY2-CRY2 interactions. The C-terminal region is predominantly negatively charged, with only a small region containing two positively charged residues Arg487 and Arg489. The mutational analysis indicated that the positively charged residues enhance the oligomerization; however, negatively-charged residues counterbalanced this effect. The proposed deduction explained why the Glu490Gly substitution (discussed above) [44] improved the formation of CRY2 clusters. The variants with positively charged residues substitutions (Glu490Arg or Glu490His) exhibited even more improved clustering capacities than Glu490Gly. Based on these findings, the authors engineered two variants designated as CRY2high, with enhanced clustering containing the positively charged substitution Glu490Arg, and CRY2low with reduced oligomerization by using the negatively charged mutations CRY2 (1-488) EED. The variants were demonstrated to enable different extents of extracellular signal-regulated kinase (ERK) activation by optogenetic manipulation. Despite the efforts in uncovering the mechanism of CRY2 clustering, the oligomerization interaction remains unclear and it seems increasingly likely that it involves multiple binding sites as well as stabilization by surfaces that are unchanged in both the light and dark states [120,124].

An alternative strategy to engineering of improved CRY2 variants is to improve the cluster formation by combining both CRY2-CRY2 oligomerization and CRY2-CIB1 dimerization. This idea was firstly realized in a system named LARIAT (light-activated reversible inhibition by assembled trap) described by Lee et al. in 2014 [17] (Figure 4). In the LARIAT system, the CIB1 domain is fused to a multimeric protein (MP). Light-dependent homo-oligomerization and heterodimerization induce interactions among the MPs that facilitate cluster formation. This cluster formation leads to inactivation of the biofunctional module due to becoming effectively “trapped” in the clusters. The LARIAT system was demonstrated to be useful for reversible inhibition of biofunctional domains, including Vav2 which induces membrane protrusion, PI3K which is involved in lipid metabolism, and microtubule polymerization in formation of mitotic spindle structure which is crucial in mitosis. A similar strategy has also been used to detect the protein interactions via visualization of the colocalization of the fluorescent clusters and their binding partners [44]. Recently, this design strategy has been expanded by combining LARIAT with RNA-binding proteins [125]. The resulting mRNA-LARIAT system was demonstrated to manipulate the localization and translation of specific mRNAs in cells.

Overall, the engineering of CRY2 has been somewhat limited due to our current lack of understanding of the light-dependent interactions at the molecular level. There is no reported crystal structure for *Arabidopsis* CRY2 available at the time of writing this review. However, CRY2-based optogenetic tools have proven effective for providing a spatiotemporal control via a variety of design strategies.

## 5. Phytochromes-Based Optogenetic Tools

Phytochromes are a class of photoreceptors found in plant, fungi, and bacteria which contain bilin chromophores and are sensitive to red and far-red light. The proteins in this superfamily consist a highly conserved photosensory region, and a diversified dimerization or signaling region [126,127]. The photosensory region typically is composed of three adjacent domains: a PAS domain, a cGMP phosphodiesterase/adenylyl cyclase/FhlA (GAF) domain and a phytochrome-specific (PHY) domain. The chromophore binding occurs in the pocket of GAF domain, with the bilin cofactor covalently connected to a conserved Cys residue located in the GAF or PAS domain, depending on the particular species [127]. For the prototypical phytochrome, the photosensory domain exists in a stable red-light-absorbing conformation (Pr) in the dark state. Upon absorption of red light, Pr reversibly converts into a far-red-absorbing form (Pfr). Pfr can be converted back to Pr by illumination with far-red light or through a dark recovery [128]. The red-shifted wavelengths absorbed by phytochromes (from 620 to 800 nm) [127] are mostly located in the near-infrared (NIR) window [129]. Within the NIR window, the absorption of water molecules and hemoglobin in blood is reduced, therefore light has the maximal depth of penetration in biological tissues [129]. In addition, red light and far-red light illumination minimizes the cross-talk with blue light excitation used for green fluorescent indicators such as GCaMP [130]. Due to their NIR absorbance, phytochromes represent an advantageous template for development into both optogenetic actuators [25,54,56] and fluorescent indicators [131,132,133,134]. Here we review the toolkit of phytochrome-based actuators derived from plant phytochromes and bacterial phytochromes.

### 5.1. Engineering of Plant Phytochrome-Based Actuators

The plant phytochromes are light-sensitive proteins responsible for regulating various plant biological processes, including growth, germination, flowering, and circadian rhythm [90,126,135]. However, only *Arabidopsis* phytochrome B (PhyB) [136,137,138] has been employed as a photosensory module in optogenetic actuators. Upon red or far-red light illumination, the protein conformational change is initiated by the isomerization of phycocyanobilin (PCB) or phytochromobilin (PΦB) chromophore [139] which is endogenous in plant and cyanobacteria but is not present in mammalian cells.

With *cis* to *trans* isomerization, PhyB converts from an inactive Pr form into an active Pfr form, which in turn binds to the phytochrome-interacting factor (PIF). The PhyB-PIF pair was first demonstrated to be useful as an optogenetic tool in mammalian cells by Levskaya et al., in 2009 [54] ((ii) in Figure 1). To utilize this protein pair as actuators, they firstly confirmed that PhyB could covalently bind to the exogenously added PCB chromophore in mammalian cells. They then optimized PhyB (photosensory region with the C-terminal tandem PAS domains) and PIF (N-terminus of PIF6) by screening of variants using a protein translocation assay. The improved “PhyB-PIF” pair exhibited reversible light-dependent association and dissociation both within a few seconds (τ_on_ = 1.3 s and τ_off_ = 4 s). Compared to other light-induced dimerization system such as VVD and CRY2-CIBN pair, the controllability, fast kinetics of both association and dissociation, and red-shifted illumination of PhyB-PIF are major advantages of this system. With these advantages, Phy-PIF pair has enabled spatial and temporal control with high resolution in various cell biology applications via light-controllable PPI and protein translocation [54,55,140,141,142,143]. Examples include the optical control of Rac1 and Cdc42-mediacted G-protein signaling [54], gene expression by activation of transcriptional factors [140,141], and Cre-mediated DNA recombination in yeast cells [143].

Despite the unique advantages of PhyB-PIF system, the major drawback of this system is that the PCB chromophore is only produced in photosynthetic organisms, thus PCB needs to be added externally when using Phy-PIF in mammalian cells. In 2017 Uda et al. reported overcoming this limitation by reconstructing the PCB biosynthesis pathway in bacterial and mammalian cells [144]. They reported that endogenous heme can be converted into PCB by expressing a synthetic gene construct “PHFF” encoding four proteins: Heme oxygenase (HO1), ferredoxin (Fd), Fd-NADP+ reductase (Fnr), and phycocyanobilin:ferredoxin oxidoreductase (PcyA) [145,146,147]. The amount of PCB produced via biosynthesis was sufficient to reconstitute the light-dependent Phy-PIF interactions in cell assays [144].

Overall, the applications of plant phytochrome-based photosensory modules in optogenetic actuators are rather limited to date. Although the PCB biosynthesis pathway has been transplanted into mammalian cells, this strategy is not ideal due to the extra steps required to incorporate the biosynthesis related genes into the targeted cells. The efficiency of PCB production can also be impacted by the metabolism of intermediates [144]. One potential solution to solve these problems, is engineering the chromophore binding pocket from binding PCB to binding biliverdin (endogenous in mammalian cells) via rational design guided by the PhyB crystal structure [148].

### 5.2. Engineering of Bacterial Phytochrome-Based Actuators

Similar to plant phytochromes, bacterial phytochrome photoreceptors (BphPs) undergo photoisomerization between Pr and Pfr forms upon illumination with red or far-red light. However, unlike the plant phytochromes, some bacterial phytochromes adopt the Pfr form, rather than the Pr form, in the ground state [149]. Another difference relative to plant phytochromes is that some eubacteria phytochromes use biliverdin IXα (BV) tetrapyrrole as the cofactor chromophore [127,150]. As a product of heme catabolism, BV also exists ubiquitously in mammalian cells [151], therefore bacterial phytochromes are more feasible to be used as photosensory modules in optogenetic tools. Notably, bacterial phytochrome has been extensively engineered into various fluorescent reporters that have been utilized widely for cell imaging (such as iRFP variants [132,152]), a protein interactions reporter (Split [131,132]), and a genetically encoded calcium indicator (NIR-GECO [133,134]).

The first example of a bacterial phytochrome-based optogenetic actuator, based on bacterial phytochrome BphP1 and a natural gene expression regulator PpsR2, was reported by Kaberniuk et al. in 2016 [25] ((ii) in Figure 1). PpsR2 and the Pfr form of BphP1 and are both homodimers in dark state. Absorption of far-red light (~740 nm) by BV chromophore in BphP1 disrupts the homodimerization and induces heterodimerization between BphP1 and PspR2 [25,153]. In 2017, Redchuk et al. optimized PspR2 variant by identifying the minimal domain required for interaction which is termed as Q-PAS1 [56]. Using this optimized variant, light-dependent interactions were demonstrated via protein translocation assay in cells [25,56] and in vivo light-induced gene expression in mice upon illumination of far-red light [25]. Recently, the utility and versatility of the BphP1-Q-PAS1 pair has been further demonstrated in combination with intrabodies [154]. The resulting light-controllable intrabodies enable multiplex protein regulation of endogenous protein (actin and RAS GTPase) with spatiotemporal precision.

However, one of the limitations of the BphP1-PspR2 system is the poor controllability of the dissociation step. Although the red light (~650 nm) illumination accelerates BphP1-PspR2 dissociation, the complete dissociation of the heterodimer only occurs through dark state relaxation. The relative slow dark recovery rate in cell experiments (*t*_1/2_ ~900 s) [25] suggests there is further room for improvement.

## 6. Fluorescent Protein-Based Actuators

The discovery of the *Aequorea victoria* green fluorescent protein (avGFP) [155,156,157] ultimately launched a new era in cell biology studies. AvGFP enables researchers to monitor biological processes in living cells and tissues by genetically fusing the FP to the protein of interest [158,159] or utilizing any of the numerous biosensors based on engineered FPs [160,161,162]. AvGFP does not require an exogenous cofactor as the chromophore is generated from the autogenic post-translational modification of three amino acids with the polypeptide sequence (Ser65, Tyr66 and Gly67 in avGFP) [160]. Taking avGFP as an example, it is generally believed that the mechanism of chromophore formation involves cyclization, oxidation, and dehydration steps [163]. The mature chromophore is covalently bound with a central α-helix surrounded by a highly conserved β-barrel.

In addition to emitting fluorescence upon excitation, the chromophores in some FP variants also undergo photochemical reactions. These FPs are referred to as phototransformable FP (PTFPs) [164]. In contrast to the conventional FPs which exhibit constant fluorescence, the intensity or emission wavelength of PTFPs typically changes upon light illumination. Based on their different light-induced responses, the PTFPs can be divided into three main classes: (i) Photoactivatable FP [165,166,167], (ii) photoconvertible FP [168,169,170], and (iii) photoswitchable FP [171,172,173]. The photochemistry of chromophores in PTFPs has expanded the FP applications in important directions (e.g., super-resolution imaging), and has inspired researchers to engineer FPs into photosensory modules for optogenetic actuators.

### 6.1. Engineering of Dronpa-Based Actuators

For photoswitchable FP, the reversible photoswitching is achieved through photoisomerization of the chromophore [174,175]. The protein can be switched between the dark state and fluorescent state by illuminating with two distinct wavelengths of light. The photochemistry is typically accompanied by a rearrangement of the chromophore pockets [175,176,177]. For example, in the photoswitchable FP Dronpa [171], the β-strands adjacent to chromophore were observed to be destabilized (by NMR analysis) [176] with cyan light illumination. That is, the *cis* to *trans* isomerization causes partial unfolding of the β-barrel and disturbs the interaction on cross-dimer interface in the tetrameric parent [176]. Taking advantage of this light-induced conformational change, in 2012 Zhou et al. [46] reported the optogenetic control of protein activity by using a previously reported Dronpa mutant (Dronpa145N) [178]. Illumination with cyan light induces the tetramer to monomer conversion of Dronpa145N, or the dimer to monomer conversion of the Dronpa145K variant [46]. Although the affinity of Dronpa variants in multimeric form is relatively low, this system is suitable for “intramolecular oligomerization”-based applications by fusing two copies of Dronpa ((ii) in Figure 1). The strategy was utilized for optical control of biofunctional domains such as Cdc42 GEF intersectin, HCV protease, and various kinases [32,46].

### 6.2. Engineering of PhoCl-Based Actuators

Another PTFP-based photosensory domain for actuators is the photocleavable protein (PhoCl) [58], engineered from the green-to-red photoconvertible FP, mMaple [179]. This class of protein contains a chromophore that derives from the tripeptide His-Tyr-Gly. The photoconversion reaction is a violet light-induced β-elimination reaction, which extends the conjugated system of the green chromophore to create a red chromophore [180,181]. In mMaple, the bond cleavage of the polypeptide backbone forms a ~66 residues N-terminal fragment and a ~166 residues C-terminal fragment that remain associated. A topological variant of mMaple, named “PhoCl”, was reported by Zhang et al. in 2017 [58]. To achieve the light-induced dissociation, PhoCl was engineered via circular permutation, followed with screening for variants with loss of red fluorescence after conversion. Upon violet light illumination, cleavage of PhoCl produces a ~10 residues small C-terminal peptide fragment and a ~220 residues large “empty barrel” N-terminal fragment which can dissociate spontaneously.

Although the advantages of the synthetic photocleavable linkers have been widely recognized in chemical biology research [182,183,184], PhoCl represents the first case of genetically encodable photocleavable tag that is composed of only natural amino acids. In previously reported study, PhoCl1 has been utilized for optical control of the biofunctional modules through three strategies (Figure 5).

The first of these three strategies ((i) in Figure 5) involves using PhoCl to modulate the activity of biofunctional modules through manipulation of protein localization. In this approach, PhoCl and the biofunctional module are fused to a localization tag (such as nuclear exclusion sequence (NES) [185], nuclear localization sequence (NLS) [20], and membrane targeting motif (CAAX) [186]), such that the function of the biofunctional domain is caged in the dark state due to the inaccessibility to its target area. After transient illumination, subcellular protein translocation is manipulated by optical detachment of the localization tag. Another strategy ((ii) in Figure 5) is to design the PhoCl-mediated actuators based on steric controls. Steric effect was introduced to the biofunctional module by fusing to steroid receptor (SR) domains [187]. Caging of the protein of interest likely involves interaction between SR domain with heat shock protein 90 (Hsp90), which leads to steric blocking, partial unfolding, and sequestering the biofunctional domain in cytoplasm. Compared to the translocation strategy, the steric control has enabled the activation of protein with much lower background signals. The third design strategy ((iii) in Figure 5) is to utilize PhoCl as a photocleavable linker between the biofunctional domain and its proteinaceous inhibitor. In this generic approach, the protein function would be inhibited in the dark state, until the inhibitor is released by photocleavage. Based on these designs, PhoCl was successfully employed for engineering of photoactivatable Cre recombinase, photoactivatable Gal4 transcription factor, and light-activation of protease-activatable ion channel pannexin-1 (PanX1).

## 7. Conclusions

Engineering of photosensory domains facilitates the development of optogenetic actuators with diversity in terms of mechanism, spatiotemporal scale, and applicability to control of various processes. The burgeoning repertoire of photosensory building blocks has enabled the optical control of a variety of well-defined biological processes. Yet the demand for photosensory domains with new capabilities, such as optimal kinetics for specific biological events, shifted stimulation wavelengths to minimize crosstalk with other optical tools, and compatibility with two-photon microscopy to achieve precise targeting deeper within tissue, is ever-growing. Bioinformatic approaches may yet reveal new classes of photosensitive protein from nature with desirable new features. Meanwhile, established protein engineering approaches including directed evolution, rational protein design, and computational modeling, will continue to provide new variants with improved capabilities. Currently, engineering of the naturally sourced photoreceptors is still limited due to several reasons. First, few atomic-resolution structures exist for both dark state light state of the photo-sensitive proteins. The light-dependent conformational changes are unlikely to be revealed by conventional protein X-ray crystallography, due to the compact nature of crystal packing and the stabilizing effects of interprotein contacts. Further structural studies by using protein NMR, cryo-EM analysis and computational simulations may help researchers to better understand mechanisms of these light-induced responses at the molecular level. Second, high throughput screening methods are required for actuator engineering. In previous studies, most of the improved variants were identified by rational design and mutational analysis. Exploring the suitable high through-put screening strategy, according to the specific application, may provide a robust method to identify variants with more substantial improvements.

## Figures and Tables

**Figure 1 ijms-21-06522-f001:**
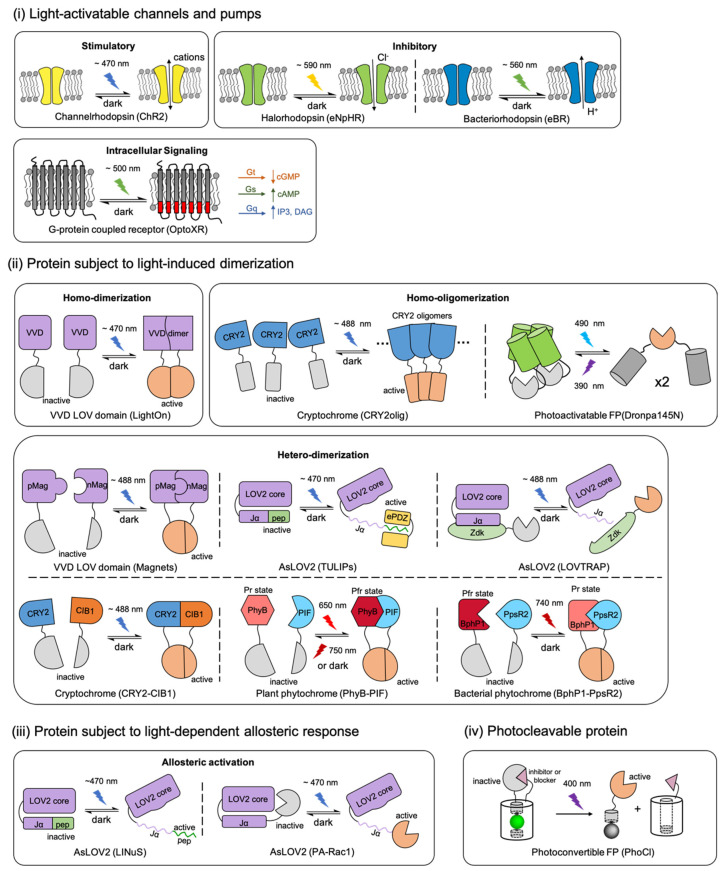
A schematic overview of the photosensory modules in actuators and their representative applications. Proteins are classified by different light-induced responses and labeled with their natural sourced photoreceptors. (**i**) Opsin-based light-activatable transmembrane protein include microbial rhodopsins and G-protein couple receptor (GPCR) rhodopsins. (**ii**) Light-induced homo-dimerization, homo-oligomerization, or hetero-dimerization is enabled by the interactions of light-oxygen-voltage-sensing domain (LOV) domains, cytochromes, phytochromes or photoswitchable fluorescent protein (FP) with their binding partners. (**iii**) Activity of the biofunctional module in actuator can be modified by light-induced allosteric response of LOV domains. (**iv**) Photocleavable protein (PhoCl) engineered from a photoconvertible FP enables the covalent caging of biofunctional module that can be activated irreversibly by light. The schematic representations in this figure are inspired and adapted from previously reported studies, which are labelled in parentheses below the corresponding schematic. The citations are as follow: ChR2 [2,34,35]; eNpHR [2,34,36]; eBR [34,37]; OptoXR [2,34,38]; LightOn [24]; CRY2oligo [44,45]; Dronpa145N [46]; Magnets [18,47,48]; TULIPs [49]; LOVTRAP [50,51,52]; CRY2-CIB1 [27,28,53]; PhyB-PIF [54,55]; BphP1-PpsR2 [25,56]; LINuS [20,21]; PA-Rac1 [57]; and PhoCl [58].

**Figure 2 ijms-21-06522-f002:**
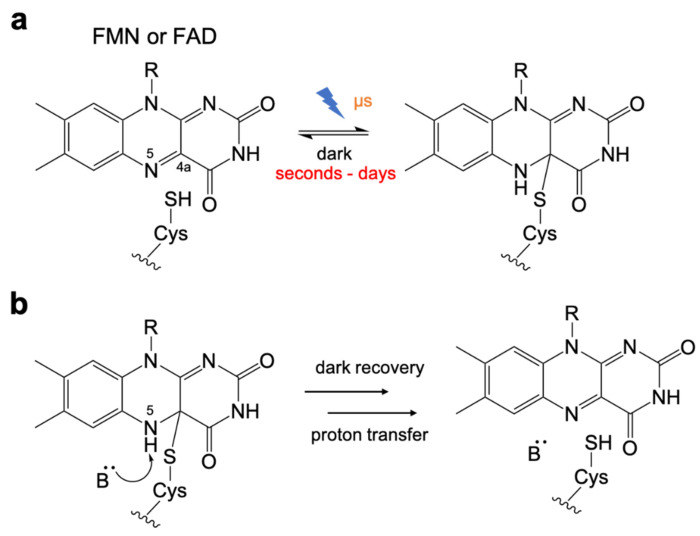
Photochemistry of LOV domain. (**a**) The photochemistry of flavin chromophore. Upon illumination, a reversible thioether bond is formed between the flavin C4a position and the thiol of cysteine. (**b**) Base-catalyzed cleavage for dark recovery. Cys-flavin adduct cleavage process is assisted by deprotonation of the flavin N5 position.

**Figure 3 ijms-21-06522-f003:**
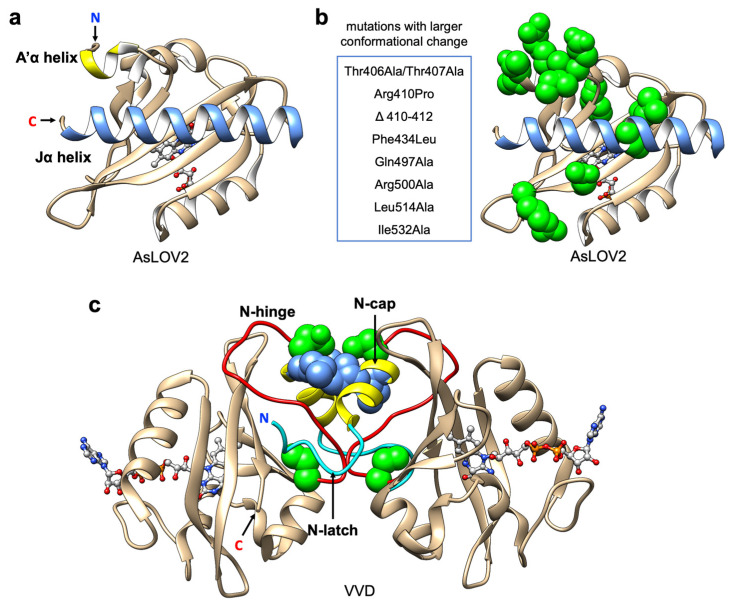
LOV domain structure and protein engineering. (**a**) Representative structure of AsLOV2 domain (PDB ID: 2V1B). Jα-helix is represented in blue; A’α helix is represented in yellow; flavin cofactor (FMN) chromophore is represented with light grey ball and stick; (**b**) Mutations that increase conformational change in AsLOV2 domain. Mutations listed are represented with green spheres. (**c**) Light-state dimer of Vivid (VVD) domains (PDB ID: 3RH8). N-Cap helixes are represented in yellow; N-hinge loops are represented in red; N-latch peptides are represented in cyan; FAD chromophores are represented with light grey ball and stick; mutations (Cys71Val and Asn56Lys) that increase the dimer affinity are represented with green spheres; mutations (Ile52Arg and Met55Arg in pMag; Ile52Asp and Met55Gly in nMag) that used in Magnet system are represented with blue spheres.

**Figure 4 ijms-21-06522-f004:**
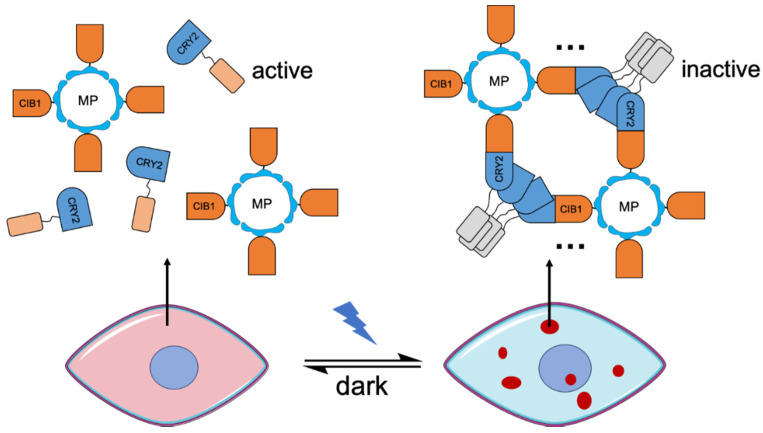
A schematic of the LARIAT (light-activated reversible inhibition by assembled trap) system. Inhibition of biofunctional modules based on a combination of cryptochrome 2 (CRY2) homo-oligomerization and CRY2-CIB heterodimerization. Light-induced clusters are represented in dark-red dots. MP, multimeric protein.

**Figure 5 ijms-21-06522-f005:**
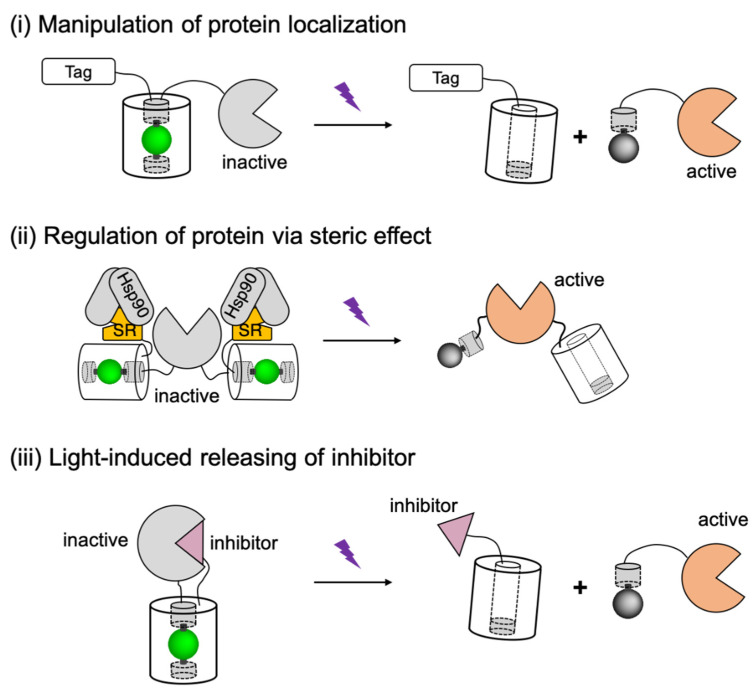
Schematic representations of PhoCl-based actuators design strategies. Tag, localization tag; Hsp90, heat shock protein 90; SR, steroid receptor. The schematic figure is adapted from the reported literature [58].

**Table 1 ijms-21-06522-t001:** Dark recovery kinetics of AsLOV2 variants.

Protein Variants	Kinetics: τ_FMN_ (s)	Reference
WT	55	Kawano et al., 2013 [93]
	68.3	Nash et al., 2008 [83]
	80	Zayner and Sosnick., 2014 [87] Zayner et al., 2012 [88]
	81	Zoltowski et al., 2009 [94]
Asn414Asp	69	Zayner and Sosnick., 2014 [87]
Asn414Gln	280	
Asn414Gly	615	
Asn414Ser	685	
Asn414Thr	892	
Asn414Ala	1427	
Asn414Leu	1847	
Asn414Val	≥12 h	Zayner et al., 2012 [88]
Val416Ile	821	Zoltowski et al., 2009 [94]
Val416Ile/Leu496Ile	1009	
Val416Thr	2.6	Kawano et al., 2013 [93]
Val416Leu	4300	
Ile427Val	4	Kawano et al., 2013 [93]
Phe434Leu	12	Zayner et al., 2012 [88]
Cys450Val	NM ^1^	Zayner and Sosnick., 2014 [87]
Leu453Val	160	Zayner and Sosnick., 2014 [87]
Phe494Leu	206	Zayner et al., 2012 [88]
Phe494Cys	282	Zayner and Sosnick., 2014 [87]
Phe494His	NM	
Gln513Asn	37	Nash et al., 2008 [83]
Gln513Leu	1080	
Gln513Ala	261	Zayner et al., 2012 [88]
Gln513Asp	5	Zayner and Sosnick., 2014 [87]
Gln513His	30	
Gln513Leu	1793	
Asn414Ala/Gln513His	2	Zayner and Sosnick., 2014 [87]
Asn414Leu/Gln513Ala	1900	
Asn414Ala/Gln513Ala	2081	

^1^: NM, not measurable.

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
