# Peer review of "Engineering Photosensory Modules of Non-Opsin-Based Optogenetic Actuators"

_ijms, 2020, doi:10.3390/ijms21186522_

Round 1

Reviewer 1 Report

The Authors present a really nice Review of the currently available strategies for engineering optogenetic controllers of biological activity within cells. In particular the Review focuses on the properties and mechanisms of function of different classes of optogenetic actuators based on i) LOV domain, ii) Cryptochromes, iii) Phytochromes, and iv) fluorescent proteins.

Comments:

>please revise Figures with the following:The symbols of thunderbolts hide important information related to the wavelenghts for activation/inactivation of each optogenetic molecule. 

>Make it clear that for ChR2 the transmembrane flow of ions can be bidirectional (use a double headed arrow), since this is a channel, rather than an unidirectional pump.

>Line 198: the Authors should define what they intend for 'dynamic range' in this specific context.

>Line 275: the authors should mention also the optical control of nanobody activity that was achieved using the pMAG VVD system (Daseuli Yu et al., Nat Methods 2019).

>Conclusion could be expanded to discuss what are the prospects of expanding the class of protein-based photoactive molecules by natural sources.

Author Response

Reviewer #1:

The Authors present a really nice Review of the currently available strategies for engineering optogenetic controllers of biological activity within cells. In particular the Review focuses on the properties and mechanisms of function of different classes of optogenetic actuators based on i) LOV domain, ii) Cryptochromes, iii) Phytochromes, and iv) fluorescent proteins.

Response: We thank you for the supportive comments on our review. We've edited our manuscript according to the reviewers' comment. Please see the coverletter in the attachment, it details all the revisions that have been made. 

Comments:

Point 1: >please revise Figures with the following: The symbols of thunderbolts hide important information related to the wavelenghts for activation/inactivation of each optogenetic molecule. 

Response 1: Thank you for point this out. This likely due to the file conversion to pdf. To avoid this problem, we now have removed all the glow effects in the figures (Figure 1,2,4,5).

Point 2: >Make it clear that for ChR2 the transmembrane flow of ions can be bidirectional (use a double headed arrow), since this is a channel, rather than a unidirectional pump.

Response 2: We now have used bidirectional arrow in schematic of ChR2 ((i) in Figure 1).

Point 3: >Line 198: the Authors should define what they intend for 'dynamic range' in this specific context.

Response 3: Instead of using the vague term “dynamic range”, we now have replaced with more specific description: “screening for variants with low dark state activity and high protein translocation efficiency upon illumination” (line 200-201).

Point 4: >Line 275: the authors should mention also the optical control of nanobody activity that was achieved using the pMAG VVD system (Daseuli Yu et al., Nat Methods 2019).

Response 4: We have cited the work of the optical control of intracellular antibodies in line 278: “In particular, light-induced heterodimerization provides a simple and versatile way to control the activity of split enzymes including Cre recombinase [109], CRISPR-Cas9 [18], T7 RNA polymerase [60] and intracellular antibodies (intrabodies) [110].

Point 5: >Conclusion could be expanded to discuss what are the prospects of expanding the class of protein-based photoactive molecules by natural sources.

Response 5: Thanks for your comment. We now have edited the conclusion paragraph and added a brief discussion on the prospects in line 552: “Yet the demand for photosensory domains with new capabilities, such as optimal kinetics for specific biological events, shifted stimulation wavelengths to minimize crosstalk with other optical tools, and compatibility with two-photon microscopy to achieve precise targeting deeper within tissue, is ever-growing. Bioinformatic approaches may yet reveal new classes of photosensitive protein from nature with desirable new features. Meanwhile, established protein engineering approaches including directed evolution, rational protein design, and computational modeling, will continue to provide new variants with improved capabilities.”

Reviewer 2 Report

This is an excellent review on optogenetic instruments. It is comprehensive and nicely written and illustrated. An important feature of this review is detailed discussion of structure and mutagenesis of optogenetic proteins. It will be of high interest for both developers and end users of these promising technologies. I have a few minor remarks only:

1. In the Introduction, the authors stated that “we aim to provide a timely overview of the engineering of non-opsin-based photosensory proteins and their representative applications in subcellular levels”. Thus, I would suggest making the title more focused accordingly, e.g.: “Engineering photosensory modules of non-opsin optogenetic actuators”

2. Some optogenetic proteins are temperature-sensitive. For example, the heterodimerization system based on pMag and nMag requires incubation of cells at room temperature to be active. Please add discussion of this issue, as it is a significant consideration for mammalian models.

3. In the PDF version available for review, there is improper representation of “lightnings” for light illumination – probably a result of poor file conversion for image layers with gradient colors.

Author Response

Reviewer #2:

This is an excellent review on optogenetic instruments. It is comprehensive and nicely written and illustrated. An important feature of this review is detailed discussion of structure and mutagenesis of optogenetic proteins. It will be of high interest for both developers and end users of these promising technologies.

We highly appreciate your positive comments. We've edited the manuscript according to the reviewers' comments. Please see the coverletter in the attachment, it details all the revisions have been made. 

I have a few minor remarks only:

  1. In the Introduction, the authors stated that “we aim to provide a timely overview of the engineering of non-opsin-based photosensory proteins and their representative applications in subcellular levels”. Thus, I would suggest making the title more focused accordingly, e.g.: “Engineering photosensory modules of non-opsin optogenetic actuators”

Response 1: We now have changed the title to “Engineering photosensory modules of non-opsin-based optogenetic actuators” (line 2-3) as suggested.

  1. Some optogenetic proteins are temperature-sensitive. For example, the heterodimerization system based on pMag and nMag requires incubation of cells at room temperature to be active. Please add discussion of this issue, as it is a significant consideration for mammalian models.

Response 2: We now added a discussion on the temperature sensitive issue of the VVD-based system in line 278-282: “Notably, the VVD-based optogenetic tools have been reported to be temperature-sensitive [111,112] and stimulating cells with colder temperatures can also induce dimerization [113]. Accordingly, the pre-incubation of cells at the experimental temperature [54], and keeping the temperature constant during cell imaging, are important considerations for this system.”

  1. In the PDF version available for review, there is improper representation of “lightnings” for light illumination – probably a result of poor file conversion for image layers with gradient colors.

Response 3: Reviewer 1 also noticed this problem. Thanks for point this out. To avoid this problem, we have removed all the glow effects in the figures (Figure 1,2,4,5).
